# Robust Learning of Tractable Probabilistic Models

**Rohith Peddi**[1]          **Tahrima Rahman**[1]          **Vibhav Gogate**[1]

[1]The University of Texas at Dallas

## Abstract

Tractable probabilistic models (TPMs) compactly represent a joint probability distribution over a large number of random variables and admit polynomial time computation of (1) exact likelihoods; (2) marginal probability distributions over a small subset of variables given evidence; and (3) in some cases most probable explanations over all non-observed variables given observations. In this paper, we leverage these tractability properties to solve the *robust* maximum likelihood parameter estimation task in TPMs under the assumption that a TPM structure and complete training data is provided as input. Specifically, we show that TPMs learned by optimizing the likelihood perform poorly when data is subject to adversarial attacks/noise/perturbations/corruption and we can address this issue by optimizing robust likelihood. To this end, we develop an efficient approach for constructing uncertainty sets that model data corruption in TPMs and derive an efficient gradient-based local search method for learning TPMs that are robust against these uncertainty sets. We empirically demonstrate the efficacy of our proposed approach on a collection of benchmark datasets.

## 1 INTRODUCTION

The last decade has witnessed rapid advances in deep generative models that effectively capture probability distributions over high dimensional data such as Autoregressive models (ARNs) [Larochelle and Murray, 2011], Normalizing flows [Papamakarios et al., 2021], Variational Autoencoders (VAEs) [Kingma and Welling, 2014], Diffusion based models [Sohl-Dickstein et al., 2015], and Generative Adversarial Networks (GANs). Despite their striking success in learning representations over high dimensional data, these models are severely limited in their inference capabilities, and can only answer very few inference queries in polynomial time.

Simultaneously, the field of tractable probabilistic models (TPMs) which encompasses probabilistic models that guarantee efficient computation of probabilistic inference queries has witnessed significant traction. A unified framework called Probabilistic Circuits (PCs) that includes all the tractable models such as Sum-Product Networks (SPNs) [Poon and Domingos, 2011], Arithmetic circuits (ACS), Cutset Networks (CNets) [Rahman et al., 2014], Probabilistic Sentential Decision Diagrams (PSDDs) [Kisa et al., 2014] has been developed. With memory-efficient computation variants of probabilistic circuits such as Einsum Networks [Peharz et al., 2020], the expressivity of these models has significantly increased.

Although the robustness of probabilistic models has been assessed in the context of deep generative models, it has never been evaluated in the context of tractable probabilistic models. Therefore, in this paper, we analyze the robustness of tractable models in a generative setting through the lens of robust optimization [1]. Tractable models learn to approximate the data generating distribution via maximum likelihood estimation of the model's parameters. Maximum likelihood estimation demands the data be free from corruptions. But, in the real-world, data is subjected to corruptions from a wide variety of sources such as measurement errors, adversaries, and noise. The goal of this paper is to learn tractable models that are immunized against these corruptions.

Robust Optimization (RO) [Ben-Tal et al., 2009] is a learning paradigm that captures data uncertainty without using probability distributions. The problems considered here are max-min variants of learning problems formulated using stochastic optimization. These max-min formulations have roots in Game theory and can be perceived as a game between an adversary who affects the available data by inducing corruption and an optimizer who reacts to this worst-case selection of the data. In this approach, we assume the

---

[1]code: https://github.com/utd-star-ai-ml/ro_tpm_uai_2022

*Accepted for the 38$^{th}$ Conference on Uncertainty in Artificial Intelligence* (UAI 2022).

presence of point-wise adversaries whose corruptions can be confined in deterministic uncertainty sets and estimate the best solution for the worst-case realization of the data. Thus RO is more conservative than stochastic optimization and establishes a sense of *being on the safe side.*

In general, robust optimization variants of tractable stochastic optimization problems may not be tractable. The choice of uncertainty sets (1) plays a vital role in determining the tractability of the problem, (2) provides the designer with the flexibility to choose a trade-off between robustness and performance, and (3) determines the similarity of the solutions obtained for a stochastic optimization problem and its robust optimization variant. Any prior knowledge about the stochastic nature of the uncertainty in data can help choose the uncertainty sets. We note that robust optimization aims to estimate fixed solutions that ensure feasibility independent of data corruptions and is different from sensitivity analysis which is typically used as a post-optimization tool.

**Contributions.** In this paper, we propose two approaches for efficiently learning tractable models immunized against measurement errors, adversarial perturbations and noise.

- In the first approach, we formulate the learning objective using the robust optimization framework. Here, we maximize the likelihood of data subject to all corruptions belonging to a constrained uncertainty set. We propose an iterative algorithm that estimates parameters of the robust model by maximizing the worst case-likelihood obtained by the perturbed data generated by an adversary.

- In our second approach, we propose a regularizer to the maximum likelihood estimation problem that adds a *nearest neighbor bias* to the learning algorithm. We see this as an effective amalgamation of two orthogonal views of capturing the training distribution and *staying on the safe side* by optimizing for worst-case perturbation of the training distribution.

- Empirically, we evaluate the proposed approaches on Twenty benchmark datasets for density estimation task on tractable models without latent variables (Cutset Networks) and tractable models with latent variables (SPNs). Our results clearly demonstrate the striking vulnerability of maximum likelihood estimation to corruptions. They also show that our proposed approach yields TPMs that have significantly higher test set log-likelihood scores on corrupted data than TPMs learned by maximizing likelihood.

To the best of our knowledge this is the first work in learning robust tractable models. [2]

---

[2][Mauá et al., 2018] proposed Credal SPNs as an attempt to robustify SPNs, where they allow parameters of the sum product networks to vary in a closed convex set. But these models are not

## 2 NOTATION & BACKGROUND

We denote a dataset using the upper case letter $X$ and individual samples using a small case letter $x$. A dataset $X$ is an ensemble of individual data samples $x_i, i = 1, \ldots, n$, i.e., $X = [x_1, x_2, \ldots, x_n]^T$. For simplicity of exposition, we focus on binary datasets, where each sample $x_i$ is a $d$-dimensional 0/1 vector, namely $x_i \in \{0, 1\}^d$.

We denote corruptions of individual samples $x_i$ using $\Delta x_i$ where $\Delta x_i$ is a $d$-dimensional 0/1 vector (or mask), 1 indicates that the particular dimension is corrupted and 0 indicates that it is not. Given a dataset $X$, we denote an ensemble of corruptions (or masks) by $\Delta X$, where each $x_i \in X$ is associated with a mask $\Delta x_i \in \Delta X$, namely $\Delta X = [\Delta x_1, \Delta x_2, \ldots, \Delta x_n]^T$. We denote an XOR operation over two binary vectors using $\oplus$. We denote the corrupted dataset by $X \oplus \Delta X = [x_1 \oplus \Delta x_1, \ldots, x_n \oplus \Delta x_n]^T$. We denote probability mass function parameterized by $\theta$ at $x$ using $f(\theta, x)$ and log-likelihood for the dataset $X$ by $LL(\theta, X)$.

### 2.1 GENERATIVE TRACTABLE MODELS

Generative tractable probabilistic models (TPMs) such as thin junction trees [Bach and Jordan, 2001], bounded-treewidth Bayesian networks [Elidan and Gould, 2008], arithmetic circuits [Shen et al., 2016], cutset networks [Rahman et al., 2014], mixtures of cutset networks [Rahman and Gogate, 2016], and sum-product networks [Poon and Domingos, 2011] compactly represent large multi-dimensional probability distributions while ensuring that several inference and estimation tasks can be solved in time and space that scales polynomially (and often linearly) with the size of the model. TPMs may either have latent variables or they may not. Latent variables typically improve the goodness-of-fit of the models as measured by test-set log-likelihood scores while sacrificing tractability for some inference tasks such as most probable explanation. In general, inference tasks such as computing the log-likelihood, estimating marginal distribution over a subset of variables given evidence are tractable on the aforementioned TPMs while the most probable explanation task is polynomial only on TPMs having no latent variables.

In a standard setting, for generative parameter learning of tractable probabilistic models (TPMs) we seek to estimate parameters $\theta$ that maximize the log-likelihood function.

$$\max_{\theta} \sum_{i=1}^{n} \log \left( f\left( \theta; x_i \right) \right) \qquad (1)$$

In subsequent sections, we focus on two types of tractable probabilistic models, one having latent variables and the

---

scalable and unlike SPNs do not admit efficient computation of likelihoods and marginal probability distributions given observations.

second having no latent variables. We chose cutset networks (CNs) [Rahman et al., 2014] as our choice for tractable models without latent variables and SPNs [Poon and Domingos, 2011] for tractable models with latent variables, but our results can be easily applied to other tractable models. In CNs, the log-likelihood function is concave and the maximum-likelihood estimate can be computed in closed form. On SPNs, the log-likelihood function is not concave and one has to use iterative algorithms such as gradient ascent and soft/hard expectation-maximization (EM) or their stochastic versions to find parameters that correspond to a local maxima of the log-likelihood function.

We leverage the fact that in tractable models such as CNs and SPNs, given a dataset $X$ or a corrupted dataset $X \oplus \Delta X$, each parameter $\theta_i \in \theta$ can be expressed as a conditional probability and the *gradient of the log-likelihood w.r.t. $\theta_i$ can be computed in polynomial time* (cf. [Peharz et al., 2017], [Darwiche, 2009]).

## 2.2 ROBUST MAXIMUM LIKELIHOOD ESTIMATORS

We assume presence of corruptions $\Delta x_i$ which differentiates true unobserved samples $x_i^{\text{true}}$ from the observed samples $x_i^{\text{obs}}$ and motivate learning through the lens of robust optimization paradigm. Specifically, we operate under the assumption that $x_i^{\text{obs}} = x_i^{\text{true}} \oplus \Delta x_i$ i,e observed samples $x_i^{\text{obs}}$ are masked variants of true samples $x_i^{\text{true}}$ and seek to estimate parameters $\theta$ that maximize probability density of true samples.

$$\prod_{i=1}^n f\left(\theta; x_i^{\text{true}}\right) \equiv \prod_{i=1}^n f\left(\theta; x_i^{\text{obs}} \oplus \Delta x_i\right)$$

or equivalently maximize the log-likelihood function

$$LL\left(\theta; X^{\text{obs}} \oplus \Delta X\right) \equiv \log\left(\prod_{i=1}^n f\left(\theta; x_i^{\text{obs}} \oplus \Delta x_i\right)\right)$$

[Bertsimas and Nohadani, 2019] have shown that based on the modelling choice of corruptions $\Delta x_i$ and the knowledge about them, we get two types of estimators.

- **Adversarially Robust Estimators** (AREs) are obtained when we consider the corruptions reside in a deterministic uncertainty set and no further knowledge about the corruptions is available.

- **Distributionally Robust Estimators** are obtained when corruptions can be considered as random variables with known support.

# 3 APPROACH

## 3.1 UNCERTAINTY SETS

At a high level, an uncertainty set defines a boundary or region (of assignments) that is close to each observed data point $x_i^{obs}$ such that the true data point $x_i^{true}$ can be any one of the assignments in this region. We assume no prior knowledge about the corruptions and model them to reside in a deterministic uncertainty set. Specifically, we model corruptions $\Delta x_i$ to reside in an uncertainty set constrained on $L_1$ or equivalently hamming distance (since we assume binary data) and express the uncertainty set denoted by $\mathcal{U}_h$ as

$$\mathcal{U}_h = \{\Delta X = [\Delta x_1, \dots \Delta x_n]^T | \|\Delta x_i\|_1 \le h,$$
$$i = 1, \dots, n;\ h\text{-Hamming distance threshold}\}$$

We define the strength of an adversary based on the choice of uncertainty set used to corrupt the data, i.e., an adversary which can produce corruptions from an uncertainty set defined by $h = 5$ is stronger in capacity than an adversary which can produce corruptions from an uncertainty set defined by $h = 3$.

## 3.2 ADVERSARIALLY ROBUST ESTIMATORS

Roughly speaking, we define *robust log-likelihood* as the log-likelihood score of the model under the worst case realization of the data. In a robust setting, we seek to estimate parameters $\theta$ using robust maximum likelihood estimators which assume the presence of corruptions in the data $\Delta X$ and maximize the likelihood of the true samples $X \oplus \Delta X$. In the real world, we are oblivious to these corruptions and assume their presence in uncertainty set $\mathcal{U}_h$. Therefore, we seek to estimate $\theta$ that maximizes the log-likelihood against the worst-case realization of the data obtained when perturbed with corruptions $\Delta x_i$ in $\mathcal{U}$.

Formally, the robust parameter estimation task is given by

$$\max_\theta \min_{\Delta X \in \mathcal{U}_h} \sum_{i=1}^n \log\left(f\left(\theta; x_i^{\text{obs}} \oplus \Delta x_i\right)\right) \qquad (2)$$

In the above robust optimization problem (Eq. (2)), the size of the uncertainty set, which in turn depends on $h$ and $d$, determines our desire to stay on the safe side. As we increase the size of $\mathcal{U}_h$, we expect a drop in the log-likelihood score; however we immunize our our model against all corruptions from this enlarged set. We note that we solve the original maximum likelihood estimation problem (1) when $h = 0$.

Although, the max-min problem given in Eq. (2) is significantly harder in general than the traditional maximum-likelihood estimation task, it turns out that the objective

(given by the inner minimization) remains concave for cut-set networks having no latent variables.[3] This follows from the fact that the log-likelihood function is concave and minimum over a concave function is also concave. Formally,

**Proposition 1** *In CNs (having no latent variables), the optimization problem given in Eq. (2) is concave.*[4]

Thus, in CNs, since the gradient of the log-likelihood w.r.t. the parameters $\theta$ can be computed in linear time in the size of the data, the robust parameter estimation task can be solved efficiently using a sub-gradient method if the inner minimization task can be solved (optimally and) efficiently. The latter is possible when $h$ is bounded by a constant.

Unfortunately, since the log-likelihood function for SPNs is non-concave, the objective remains non-concave. For such problems, [Danskin, 1966] has shown that if the inner minimization problem can be solved optimally, then there always exists a directional derivative that can be used to update the parameters and reach a local optimum.

Formally, we can show that:

**Proposition 2** *[Danskin, 1966] Let*

$$\Delta X^*(\theta) = \arg\min_{\Delta X \in \mathcal{U}_h} \sum_{i=1}^{n} \log\left(f\left(\theta; x_i^{obs} \oplus \Delta x_i\right)\right)$$

*then*

$$\nabla_\theta \min_{\Delta X \in \mathcal{U}_h} \sum_{i=1}^{n} \log\left(f\left(\theta; x_i^{obs} \oplus \Delta x_i\right)\right)\Bigg|_{\theta=\theta_t} = $$
$$\nabla_\theta \sum_{i=1}^{n} \log\left(f\left(\theta; x_i^{obs} \oplus \Delta x_i^*(\theta_t)\right)\right)\Bigg|_{\theta=\theta_t}$$

In other words, if we can find a solution $\Delta X^*(\theta_t)$ to the inner minimization problem, then the gradient of the objective at $\theta = \theta_t$ equals the gradient of the log-likelihood of the dataset $X \oplus \Delta X^*(\theta_t)$. In SPNs, as we mentioned earlier, this gradient can be computed efficiently in time that scales linearly with the size of the model [Peharz et al., 2017].

The above discussion yields algorithm 1 where we iteratively solve the inner minimization problem to estimate corruptions $\Delta X$ from $\mathcal{U}$ and use the obtained corruptions to perturb the dataset, which shall be used to update the parameters $\theta$ of the model.

---

**Algorithm 1:** Robust Maximum Likelihood Estimation

**Input:** Binary dataset $X$, a tractable model structure having parameters $\theta$ and hamming distance threshold $h \in \mathcal{Z}$

**Output:** An assignment to $\theta$

1 **begin**
2    Randomly initialize all $\theta_i \in \theta$
3    **repeat**
     `// Solve Inner Minimization`
4      Find new set of corruptions $\Delta X$ from uncertainty set constrained by $h$ using the current parameters $\theta$
     `// Outer Maximization`
5      **for** *k steps* **do**
6        Use one step of stochastic gradient ascent or EM to update parameters $\theta$ using $X \oplus \Delta X$ (see Proposition 2)
7      **end**
8    **until** *convergence*;
9    **return** $\theta$
10 **end**

---

**Practical considerations.** Efficient estimation of adversarially robust estimators (AREs) for tractable models with latent variables depends on the efficiency and practicality of the algorithm used in finding the solution for the inner minimization problem. When exhaustive search over the space of all possible corruptions $\binom{d}{h}$ is employed in finding the optimum for the inner minimization problem, we incur a computational cost of $\mathcal{O}(d^h \times S)$ [5] (where $S$ is the size of the model). Thus, in theory, when $h$ is bounded by a constant, the optimum can be computed in polynomial time. However, exhaustive search is not practically feasible for large models (e.g., when $d > 100$ and $h > 3$). Therefore, we use a greedy local search algorithm having time complexity $\mathcal{O}(d \times h \times S)$ to search for a neighbor having the smallest log-likelihood. Since the gradient can be computed in time that scales linearly with the size of the model, when local search is employed, the overall time complexity of each iteration is reduced from $\mathcal{O}(d^h \times S \times k)$ to $\mathcal{O}(d \times h \times S \times k)$.

Using Danskin's theorem (see Proposition 2, it is straightforward to show that for SPNs and CNs, Algorithm 1 converges to a local optima of Eq. (2). In CNs, the local optima also corresponds to the global optima.

---

[3]Note that we are performing robust parameter estimation and assume that the structure of the tractable model is provided as input to our algorithm.

[4]Note that although the objective is concave, it is not smooth and therefore we have to use a sub-gradient method.

---

[5]For tractable representations that use a static ordering of variables such as algebraic decision diagrams (ADDs) and ordered binary decision diagrams (OBDDs) we can find the optimum for the inner minimization problem in time that scales polynomially with $h$, $d$ and $S$. But for dynamically ordered tractable representations such as SPNs and CNs, the time complexity of solving the inner minimization problem is exponential in $h$.

## 3.3 REGULARIZED MAXIMUM LIKELIHOOD ESTIMATORS

In a robust setting, as we increase the size of the uncertainty set (see Eq. (2)), we immunize against corruptions from a larger set and achieve better robust likelihood scores. However, these models perform poorly on the original training and test sets. To address this issue, we propose an alternative approach where we jointly optimize for both standard and robust likelihoods, weighing the latter using a regularization constant (hyperparameter) $\lambda \geq 0$.

$$
\max_{\theta} \left[ \overbrace{\left[ \sum_{i=1}^{n} \log f(\theta, x_i^{\text{obs}}) \right]}^{\text{Standard Likelihood}} + \right.
$$
$$
\left. \lambda \times \underbrace{\left[ \sum_{i=1}^{n} \min_{\Delta x_i \in \mathcal{U}} \log \left( f\left(\theta; x_i^{\text{obs}} \oplus \Delta x_i\right)\right) \right]}_{\text{Robust Likelihood}} \right] \quad (3)
$$

We can use the same algorithm (see Alg. 1) to estimate parameters $\theta$ with a minor change in Step-6 where, instead of corrupted dataset $X \oplus \Delta X$ we use augmented dataset $[X, X \oplus \Delta X]$. Roughly speaking, the optimization problem in Eq.(3) is equivalent to applying a nearest neighbor regularizer to the original (1). Our proposed approach is closely related to [Xu et al., 2009] who showed that robust linear regression under $L_\infty$ ball is equivalent to Lasso regression.

## 4 EXPERIMENTS

In this section, we evaluated the impact of our proposed parameter estimation method on both the generative and predictive performance of TPMs as well as their robustness to adversarial attacks and random noise. Our evaluation uses two popular classes of TPMs: sum product networks (SPNs) [Poon and Domingos, 2011] and cutset networks (CNs) [Rahman et al., 2014]. As mentioned earlier, we chose these two TPMs as representatives for the following two classes of TPMs: (1) TPMs having latent variables (SPNs) on which only marginal inference is tractable and (2) TPMs having no latent variables (CNs) on which both posterior marginal distributions and most probable explanations can be computed in polynomial time.

Given data, we learned both the structures and parameters of cutset networks without any latent variables using the LearnCNet algorithm proposed by [Rahman et al., 2014]. For each dataset, we initially learned a large depth cutset network and then performed a bottom-up reduce error pruning technique using the validation set to improve its generalization accuracy. Our experiments on SPNs were performed using two open-source implementations: EiNETs [Peharz et al., 2020] and RAT-SPNs [Peharz et al., 2019]. For RAT-SPNs, we used the following structural parameters for all datasets: depth $D = 3$, number of replicas $R = 50$, number

of sum nodes $C = 10$, number of input distributions $I = 10$. EiNETs use stochastic EM for estimation of parameters that maximize the likelihood of the data. We use the default parameters for online EM frequency and online EM step size (as mentioned in the author's GitHub page[6]). RAT-SPNs were trained using the DeeProb-kit[7]) library where the parameters are learnt using stochastic gradient descent with a learning rate of $1e$-2. In our experiments, we found that the performance of SPNs trained using EiNETs and RAT-SPNs are comparable across all the evaluation criteria but we noticed that the computation time of learning and inference is much faster with EiNETs. All our experiments for SPNs and CNs were performed on machine equipped with a NVIDIA A40 GPU and a 2.4 GHz Xeon 8-core processor.

For each dataset, we learned three types of SPNs and CNs: 1) SPN and CN learned by maximizing the standard data log-likelihood, 2) SPN−a and CN−a learned by maximizing robust likelihood (see Eq. (2)) of the training data, and finally 3) SPN−r and CN−r obtained by joint maximization of standard and robust likelihoods (see Eq.(3)). We performed our experiments using $\lambda = 1$. Note that the structure of all SPNs (and CNs) is learned from the original training data. The three SPNs (and CNs) differ from each other in how the parameters are learned; in other words, the structure is constant across all models. We experimented with three values, $\{1, 3, 5\}$, for the hamming distance threshold $h$. Models of types (2) and (3) were learnt on uncertainty sets $\mathcal{U}_h$ of varying size based on these hamming distance thresholds. These sets govern the size of allowable corruptions in the data.

We evaluated our method on 20 benchmark datasets that have been used in several experimental evaluations of TPMs [Lowd and Davis, 2010]. For each dataset and each $h$, we generated two additional test sets. The first test set, which we call fully adversarial test set, denoted by $\mathcal{T}_a$ was generated from $\mathcal{T}$ as follows. We begin with an empty $\mathcal{T}_a$. Then, for each test example in $\mathcal{T}$, we use *greedy local search* to find a neighbor of the example that is at most $h$ hamming distance away and has the smallest log-likelihood score w.r.t. either the SPN or CN and add it to $\mathcal{T}_a$. The second test set which we call randomly perturbed test set, denoted by $\mathcal{T}_r$, was generated from $\mathcal{T}$ as follows. We begin with an empty $\mathcal{T}_r$. Then, for each test example in $\mathcal{T}$, we select a neighbor from 100 *randomly generated neighbors* such that each neighbor is at most $h$ hamming distance away from the example and the selected neighbor has the smallest log-likelihood score w.r.t. either the SPN or CN, and add it to $\mathcal{T}_r$.

We evaluate both the generative and predictive performances of all three types of models under various corruption scenarios. To the best of our knowledge, this is the first empirical study on the robustness of expressive TPMs.

---

[6]https://github.com/cambridge-mlg/EinsumNetworks
[7]https://github.com/deeprob-org/deeprob-kit

Table 1: Generative performance: Test set log-likelihood scores of models having latent variables. $h \in \{1, 2, 3\}$: hamming distance thresholds. SPN: SPN trained original training data, SPN−a: SPN trained on the adversarially generated training data by SPN, SPN−r: SPN trained via joint maximization of standard and robust likelihoods. $\mathcal{T}$: original test data, $\mathcal{T}_a$: adversarially perturbed $\mathcal{T}$ by SPN, $\mathcal{T}_r$: randomly perturbed $\mathcal{T}$ by SPN.

| DATASET | $h$ | $\mathcal{T}$ | | | $\mathcal{T}_a$ | | | $\mathcal{T}_r$ | | |
|---|---|---|---|---|---|---|---|---|---|---|
| | | SPN | SPN−a | SPN−r | SPN | SPN−a | SPN−r | SPN | SPN−a | SPN−r |
| Plants | 1 | | -14.18 | -13.81 | -22.38 | **-18.0** | -18.31 | -22.06 | **-17.98** | -18.17 |
| | 3 | -13.56 | -16.08 | -14.61 | -39.17 | **-23.89** | -24.26 | -30.9 | **-22.91** | -23.19 |
| | 5 | | -17.88 | -14.67 | -54.85 | **-28.2** | -29.69 | -38.33 | **-26.6** | -27.25 |
| Avg. | | **-13.56** | -16.05 | -14.36 | -38.8 | **-23.36** | -24.09 | -30.43 | **-22.5** | -22.87 |
| Netflix | 1 | | -57.62 | -57.17 | -61.0 | **-59.58** | -59.92 | -60.18 | **-59.53** | -59.56 |
| | 3 | -56.84 | -59.43 | -57.72 | -67.14 | **-65.01** | -65.11 | -61.4 | -62.11 | -61.45 |
| | 5 | | -60.88 | -58.17 | -72.06 | -69.49 | **-67.69** | -62.82 | -64.24 | **-62.13** |
| Avg. | | **-56.84** | -59.31 | -57.69 | -66.73 | -64.69 | **-64.24** | -61.47 | -61.96 | **-61.05** |
| DNA | 1 | | -97.55 | -97.69 | -101.94 | **-99.47** | -99.91 | -101.18 | **-99.35** | -99.73 |
| | 3 | -97.36 | -97.73 | -97.67 | -107.32 | **-102.07** | -103.07 | -102.61 | **-100.45** | -100.88 |
| | 5 | | -98.16 | -97.6 | -111.38 | **-104.77** | -105.8 | -104.06 | **-101.83** | -101.97 |
| Avg. | | **-97.36** | -97.81 | -97.65 | -106.88 | **-102.1** | -102.93 | -102.62 | **-100.54** | -100.86 |
| Movie | 1 | | -54.21 | -54.16 | -80.05 | **-67.03** | -71.03 | -80.02 | **-67.25** | -71.03 |
| | 3 | -53.37 | -56.94 | -55.35 | -132.0 | **-85.68** | -96.0 | -104.96 | **-81.66** | -89.86 |
| | 5 | | -59.57 | -55.64 | -182.1 | **-100.87** | -122.6 | -123.16 | **-94.2** | -107.59 |
| Avg. | | **-53.37** | -56.91 | -55.05 | -131.38 | **-84.53** | -96.54 | -102.71 | **-81.04** | -89.49 |
| BBC | 1 | | **-256.59** | -272.71 | -272.79 | **-263.73** | -284.05 | -272.06 | **-263.32** | -283.35 |
| | 3 | -260.03 | **-256.49** | -263.04 | -297.09 | **-274.57** | -286.38 | -280.07 | **-268.65** | -279.18 |
| | 5 | | **-256.41** | -261.46 | -320.12 | **-282.87** | -295.06 | -287.86 | **-274.43** | -282.58 |
| Avg. | | -260.03 | **-256.5** | -265.74 | -296.67 | **-273.72** | -288.5 | -280.0 | **-268.8** | -281.7 |

## 4.1 ROBUST GENERATIVE PERFORMANCE

To evaluate the generative performance and robustness of the learned models, we compare their log-likelihood scores on three different test sets described above ($\mathcal{T}, \mathcal{T}_r, \mathcal{T}_a$) for $h = \{1, 3, 5\}$. Scores on the set $\mathcal{T}$ indicate the model's *goodness-of-fit* to the underlying data generating distribution and larger scores imply a better fit. On the other hand, scores on the sets $\mathcal{T}_a$ and $\mathcal{T}_r$ are representative of a model's robustness to adversarial and random perturbations. Higher scores imply that the model is resilient to small perturbations to the samples in $\mathcal{T}$. Tables 1 and 2 report the average log-likelihood scores of SPNs and CNs respectively obtained on $\mathcal{T}, \mathcal{T}_a$ and $\mathcal{T}_r$. For ease of readability, we only report results on five datasets with increasing dimensionality. A comprehensive set of results are provided in the supplement.

We observe that although SPNs and CNs have slightly higher scores on $\mathcal{T}$ as compared to their robust counterparts {SPN−a, SPN−r }'s and {CN−a, CN−r }'s, they have significantly lower scores on the corrupted sets $\mathcal{T}_a$ and $\mathcal{T}_r$. Both SPNs and CNs trained using our proposed approaches consistently exhibit superior robust test-set log-likelihood scores as compared with standard SPNs and CNs.

**Impact of increasing $h$:** We observe that as we increase $h$, the performance of both SPN−a and SPN−r degrades on the original test set $\mathcal{T}$, but the performance of SPN−r degrades at a slower rate than SPN−a. In particular, there is an order of magnitude difference in the likelihood scores

of SPN−a and SPN−r for $h = 5$. For cutset networks, we see the same picture; as we increase $h$, the performance of CN−r degrades at a slower rate than CN−a on $\mathcal{T}$.

Comparing between SPNs and CNs, we see that as we increase $h$, the performance of adversarial and regularized CNs degrades at a much slower rate on $\mathcal{T}$ as compared with SPNs. This slow (and more graceful) degradation is likely due to the fact that CNs are more biased and have fewer parameters than SPNs; as a result CNs are less sensitive to changes in the training data.

On the adversarial and random test sets, namely on $\mathcal{T}_a$ and $\mathcal{T}_r$ respectively, we observe that increasing $h$ significantly degrades the performance of SPNs and CNs which are trained on the original training set. For instance, there are several orders of magnitude difference between the log-likelihood scores on $\mathcal{T}_a$ (and $\mathcal{T}_r$) for $h = 5$ and $h = 1$. On the other hand, as compared with SPNs (and CNs), the rate of decrease in log-likelihoods (as we increase $h$) is much smaller for SPN−a and SPN−r (CN−a and CN−r).

**Choice of $h$:** We motivate our choice of uncertainty sets $h \in \{1, 3, 5\}$ from two viewpoints; experimental view and observational view. In our experiments, we noticed for the density estimation task, a competent adversary can easily find samples in uncertainty sets $h \in \{1, 3, 5\}$ which can bring down the log-likelihood scores by 2-3 fold and for the image completion task, an adversary can easily find samples in $h = 5$ which can completely change the output of the completed image (e.g., changing from 4 to a 9 or

Table 2: Generative performance: Test set log-likelihood scores of cutset networks or models without latent variables. $h \in \{1, 2, 3\}$: hamming distance thresholds. CN: Cutset networks trained on original training data, CN−a: CNs learned from adversarially generated training data by CNs, CN−r: trained via joint maximization of standard and robust likelihoods. $\mathcal{T}$: original test data, $\mathcal{T}_a$: adversarially perturbed $\mathcal{T}$ by CN, $\mathcal{T}_r$: randomly perturbed $\mathcal{T}$ by CN.

| Dataset | h | $\mathcal{T}$ | | | $\mathcal{T}_a$ | | | $\mathcal{T}_r$ | | |
|---|---|---|---|---|---|---|---|---|---|---|
| | | CN | CN−a | CN−r | CN | CN−a | CN−r | CN | CN−a | CN−r |
| Plants | 1 | -13.50 | -13.61 | -13.56 | -35.16 | **-29.94** | -30.68 | -25.43 | **-23.43** | -23.81 |
| | 3 | | -13.72 | -13.62 | -58.00 | **-48.74** | -49.66 | -38.97 | **-34.88** | -35.27 |
| | 5 | | -13.82 | -13.63 | -72.16 | **-58.08** | -61.65 | -49.94 | **-42.80** | -44.80 |
| Avg. | | **-13.50** | -13.72 | -13.60 | -55.11 | **-45.59** | -47.33 | -38.11 | **-33.70** | -34.63 |
| Netflix | 1 | -58.71 | -59.96 | -58.97 | -66.26 | **-62.77** | -63.59 | -62.91 | **-62.00** | -61.92 |
| | 3 | | -61.07 | -59.67 | -75.09 | **-65.83** | -67.21 | -66.56 | **-64.10** | -64.12 |
| | 5 | | -62.35 | -59.91 | -81.19 | **-67.38** | -69.92 | -69.04 | **-65.43** | -65.58 |
| Avg. | | -58.71 | -61.13 | -59.52 | -74.18 | **-65.33** | -66.91 | -66.17 | **-63.84** | -63.87 |
| DNA | 1 | -87.60 | -87.82 | -87.70 | -95.74 | **-93.52** | -93.88 | -94.37 | **-93.08** | -93.36 |
| | 3 | | -89.74 | -88.62 | -109.12 | **-99.34** | -101.06 | -103.41 | **-97.78** | -98.45 |
| | 5 | | -90.71 | -89.19 | -121.95 | **-104.54** | -107.37 | -110.50 | **-100.94** | -101.89 |
| Avg. | | **-87.60** | -89.42 | -88.50 | -108.94 | **-99.13** | -100.77 | -102.76 | **-97.27** | -97.90 |
| Each Movie | 1 | -58.20 | -58.52 | -58.21 | -124.66 | **-117.42** | -119.15 | -86.10 | **-83.96** | -84.53 |
| | 3 | | -58.70 | -58.37 | -184.36 | **-174.85** | -176.03 | -112.96 | **-109.01** | -109.62 |
| | 5 | | -58.76 | -58.77 | -233.61 | -222.43 | **-214.66** | -131.36 | -126.49 | **-125.46** |
| Avg. | | **-58.20** | -58.66 | -58.45 | -180.88 | -171.57 | **-169.95** | -110.14 | -106.49 | **-106.54** |
| BBC | 1 | -261.86 | -261.97 | -261.89 | -271.99 | **-269.79** | -270.12 | -269.98 | **-268.97** | -269.21 |
| | 3 | | -262.61 | -262.36 | -288.77 | **-278.96** | -280.94 | -277.79 | **-275.59** | -275.80 |
| | 5 | | -264.97 | -262.72 | -304.09 | **-285.92** | -290.28 | -285.14 | **-282.64** | -282.69 |
| Avg. | | **-261.86** | -263.18 | -262.32 | -288.28 | **-278.22** | -280.45 | -277.64 | **-275.73** | -275.90 |

3 to an 8 as shown in 1). For uncertainty sets $h \geq 5$, we observe that samples obtained may no longer be part of the true underlying distribution (i.e., the samples are out-of-distribution). For example, on the MNIST dataset, the difference between (0, 8), (3, 8), (4, 9), (2, 3) etc. is $\leq 3$ pixels. Similarly, the benchmark datasets used in density estimation task are curated from user click stream, page visits and preferences data; here, samples from $h \geq 5$ can completely alter the estimated distribution.

## 4.2 ROBUST PREDICTIVE PERFORMANCE

We used conditional log-likelihood (CLL) scores to evaluate the predictive performance. Given query variables $q$ and evidence variables $e$, the CLL score of a data point $x$ equals $\log f(x^q | x^e)$. We compare the average CLL scores of all models on $\mathcal{T}$, $\mathcal{T}_a$ and $\mathcal{T}_r$. We randomly selected different percentages of variables as query variables and set the remaining variables as evidence variables. The uncertainty sets are now computed over the evidence variables using greedy local search for hamming distances $\{1, 3, 5\}$.

Tables 3 and 4 report the CLL scores obtained by the various SPNs and CNs where half of the variables were set as query variables and the remaining as evidence variables. We observe a similar trend: {SPN−a, SPN−r } and {CN−a, CN−r } have better CLL scores compared to SPN and CN respectively on $\mathcal{T}_a$ and $\mathcal{T}_r$. These results demonstrate that our proposed method yields robust predictions.

Figure 1

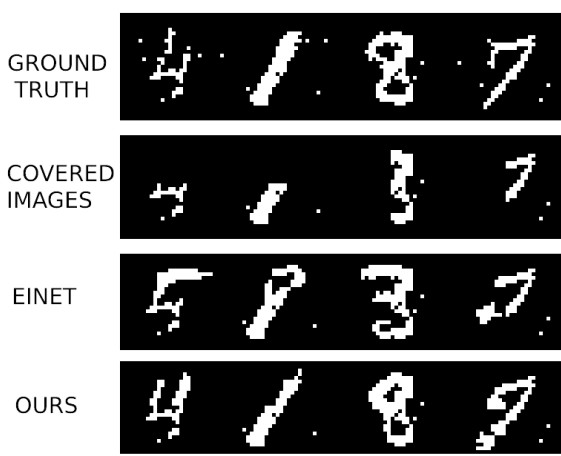

GROUND TRUTH

COVERED IMAGES

EINET

OURS

**Robust image completion:** Fig.1 shows qualitative results on the image completion task for randomly chosen images from the binarized MNIST dataset [LeCun and Cortes, 2010]. The first row shows the original corrupted images, the second row shows covered images (the top and left halves are covered in the first two and the last two columns respectively), and the third and fourth rows show reconstructions based on SPNs and robust SPNs respectively. We clearly observe that on corrupted data, robust SPNs yield better quality completions as compared to the original SPNs.

In summary, we notice that models trained on (2) pro-

Table 3: Predictive performance: Conditional log-likelihood scores given 50% evidence for models having latent variables (SPNs). $h \in \{1, 2, 3\}$: hamming distance thresholds. SPN: SPN trained original training data, SPN−a: SPN trained on the adversarially generated training data by SPN, SPN−r: SPN trained via joint maximization of standard and robust likelihoods. $\mathcal{T}$: original test data, $\mathcal{T}_a$: adversarially perturbed $\mathcal{T}$ by SPN, $\mathcal{T}_r$: randomly perturbed $\mathcal{T}$ by SPN.

| DATASET | $h$ | $\mathcal{T}$ | | | $\mathcal{T}_a$ | | | $\mathcal{T}_r$ | | |
|---|---|---|---|---|---|---|---|---|---|---|
| | | SPN | SPN−a | SPN−r | SPN | SPN−a | SPN−r | SPN | SPN−a | SPN−r |
| Plants | 1 | | -5.94 | -5.73 | -9.59 | **-7.57** | -7.95 | -9.46 | -7.8 | **-7.67** |
| | 3 | -5.64 | -7.07 | -6.21 | -16.97 | **-10.01** | -10.04 | -14.3 | **-10.49** | -10.73 |
| | 5 | | -7.96 | -6.13 | -25.6 | **-12.27** | -12.82 | -18.96 | **-12.23** | -12.72 |
| Avg. | | **-5.64** | -6.99 | -6.02 | -17.39 | **-9.95** | -10.27 | -14.24 | **-10.17** | -10.37 |
| Netflix | 1 | | -28.52 | -28.17 | -29.94 | **-29.05** | -29.07 | -29.7 | -29.28 | **-29.1** |
| | 3 | -28.02 | -29.48 | -28.51 | -32.57 | -31.26 | **-30.86** | -30.37 | -30.54 | **-29.87** |
| | 5 | | -30.53 | -28.85 | -34.92 | -33.06 | **-32.25** | -30.88 | -31.79 | **-30.64** |
| Avg. | | **-28.02** | -29.51 | -28.51 | -32.48 | -31.12 | **-30.73** | -30.32 | -30.54 | **-29.87** |
| DNA | 1 | | **-49.52** | -49.7 | -52.15 | **-50.31** | -50.66 | -51.81 | **-50.33** | -50.65 |
| | 3 | -49.58 | **-49.46** | -49.72 | -55.4 | **-51.56** | -52.16 | -52.45 | **-50.7** | -51.1 |
| | 5 | | **-49.41** | -49.47 | -57.85 | **-52.76** | -53.58 | -53.13 | **-51.34** | -51.44 |
| Avg. | | -49.58 | **-49.46** | -49.63 | -55.13 | **-51.54** | -52.13 | -52.46 | **-50.79** | -51.06 |
| Movie | 1 | | -22.82 | -22.63 | -25.47 | **-25.45** | -30.59 | -39.62 | **-31.23** | -35.59 |
| | 3 | -22.35 | -24.05 | -23.41 | -34.9 | **-30.18** | -41.66 | -54.06 | **-38.95** | -45.94 |
| | 5 | | -25.99 | -23.32 | -50.75 | **-36.12** | -54.6 | -63.07 | **-45.85** | -57.15 |
| Avg. | | **-22.35** | -24.29 | -23.12 | -37.04 | **-30.58** | -42.28 | -52.25 | **-38.68** | -46.23 |
| BBC | 1 | | **-88.78** | -99.07 | -102.32 | **-94.95** | -108.79 | -101.48 | **-94.38** | -107.99 |
| | 3 | -91.2 | **-88.83** | -92.51 | -123.22 | **-104.17** | -112.48 | -107.0 | **-97.57** | -104.67 |
| | 5 | | **-88.83** | -91.2 | -143.8 | **-111.82** | -119.87 | -112.08 | **-100.89** | -105.64 |
| Avg. | | -91.2 | **-88.81** | -94.26 | -123.11 | **-103.65** | -113.71 | -106.85 | **-97.61** | -106.1 |

duced better robust (conditional) log-likelihood scores than standard models but suffer in standard (conditional) log-likelihood scores. But, models trained on (3) have comparable robust (conditional) log-likelihoods scores to those trained on (2) and also have better standard (conditional) log-likelihood scores comparable to standard models evaluated on standard (conditional) log-likelihood scores.

## 5 CONCLUSION AND FUTURE WORK

In this paper, we presented an algorithm for learning robust Tractable Probabilistic Models (TPMs) when subjected to noise/perturbations/corruptions. At a high level, we formulate the robust learning problem as a max-min variant of the standard maximum likelihood estimation task where an adversary plays the role of a minimizer, affecting the training data by adding point-wise corruptions from a deterministic uncertainty set and the optimizer plays the role of a maximizer, learning parameters that maximize the likelihood for worst case realization of data. We develop a gradient-based local search technique for solving this max-min problem and show that because TPMs admit polynomial-time gradient computations, our algorithm converges to either a local or global optima and runs in polynomial time. Via a large experimental evaluation on standard benchmark datasets, we showed that our proposed methods perform reliably well, both in terms of generative and predictive evaluation measures, when the data is corrupted.

**Future work:** In this work, we focused on learning robust estimators using point-wise adversaries whose corruptions are confined in deterministic uncertainty sets; in future, we wish to explore learning distributionally robust estimators using stronger adversaries that can move entire observed distribution in probabilistic uncertainty sets constructed based on discrepancy measures such as Wasserstein distance, $\phi$-divergence, etc. We also wish to explore theoretical bounds for robust generalization.

**Acknowledgements**

We thank anonymous reviewers for their insightful comments which helped us significantly improve an earlier draft of this paper. Specifically, we would like to thank the anonymous reviewer who helped us formulate footnote 5. This work was supported in part by the DARPA Explainable Artificial Intelligence (XAI) Program under contract number N66001-17-2-4032, by the DARPA Perceptually-enabled Task Guidance (PTG) Program under contract number HR00112220005 and by the National Science Foundation CAREER award IIS-1652835.

**References**

Francis R. Bach and Michael I. Jordan. Thin junction trees. In Thomas G. Dietterich, Suzanna Becker, and Zoubin

Table 4: Predictive performance: Conditional log-likelihood scores given 50% evidence for models having no latent variables (CNs). $h \in \{1, 2, 3\}$: hamming distance thresholds. CN: Cutset networks trained on original training data, CN$-$a: CNs learned from adversarially generated training data by CNs, CN$-$r: trained via joint maximization of standard and robust likelihoods. $\mathcal{T}$: original test data, $\mathcal{T}_a$: adversarially perturbed $\mathcal{T}$ by CN, $\mathcal{T}_r$: randomly perturbed $\mathcal{T}$ by CN.

| Dataset | $h$ | $\mathcal{T}$ | | | $\mathcal{T}_a$ | | | $\mathcal{T}_r$ | | |
|---|---|---|---|---|---|---|---|---|---|---|
| | | CN | CN$-$a | CN$-$r | CN | CN$-$a | CN$-$r | CN | CN$-$a | CN$-$r |
| Plants | 1 | -9.61 | -9.68 | -9.64 | -31.26 | **-25.90** | -26.65 | -20.83 | **-18.87** | -19.23 |
| | 3 | | -9.80 | -9.70 | -50.67 | **-41.50** | -42.37 | -34.00 | **-29.51** | -29.91 |
| | 5 | | -9.89 | -9.72 | -61.56 | **-47.99** | -51.22 | -42.68 | **-35.87** | -37.68 |
| Avg. | | **-9.61** | -9.79 | -9.69 | -47.83 | **-38.46** | -40.08 | -32.50 | **-28.08** | -28.94 |
| Netflix | 1 | -45.29 | -46.12 | -45.39 | -52.51 | **-48.87** | -49.88 | -49.28 | **-47.95** | -48.15 |
| | 3 | | -47.03 | -45.96 | -60.77 | **-51.35** | -52.93 | -52.16 | **-49.46** | -49.71 |
| | 5 | | -48.01 | -46.13 | -65.87 | **-52.34** | -55.00 | -54.12 | **-50.58** | -50.97 |
| Avg. | | **-45.29** | -47.05 | -45.83 | -59.72 | **-50.85** | -52.60 | -51.85 | **-49.33** | -49.61 |
| DNA | 1 | -67.88 | -68.71 | -68.20 | -75.64 | **-72.17** | -72.86 | -73.68 | **-72.17** | -72.42 |
| | 3 | | -69.79 | -68.80 | -86.64 | **-77.70** | -79.34 | -80.89 | **-76.29** | -76.81 |
| | 5 | | -70.55 | -69.24 | -97.68 | **-82.01** | -84.64 | -87.61 | **-79.06** | -79.96 |
| Avg. | | **-67.88** | -69.68 | -68.75 | -86.65 | **-77.29** | -78.95 | -80.73 | **-75.84** | -76.40 |
| Movie | 1 | -41.36 | -41.50 | -41.35 | -107.57 | **-100.30** | -102.19 | -74.19 | **-71.71** | -72.49 |
| | 3 | | -41.55 | -41.38 | -159.81 | **-147.69** | -149.37 | -96.51 | **-92.43** | -93.12 |
| | 5 | | -41.65 | -41.62 | -217.39 | -203.58 | **-197.10** | -112.39 | -107.70 | **-105.93** |
| Avg. | | **-41.36** | -41.57 | -41.45 | -161.59 | -150.52 | **-149.55** | -94.36 | -90.61 | **-90.51** |
| BBC | 1 | -186.91 | -187.00 | -186.95 | -193.91 | **-193.24** | -193.32 | -193.95 | **-193.50** | -193.61 |
| | 3 | | -187.38 | -187.23 | -204.63 | **-199.52** | -200.42 | -200.05 | -199.00 | **-198.99** |
| | 5 | | -188.82 | -187.43 | -214.46 | **-204.51** | -206.93 | -205.69 | **-204.12** | -204.31 |
| Avg. | | **-186.91** | -187.73 | -187.20 | -204.33 | **-199.09** | -200.22 | -199.90 | **-198.87** | -198.97 |

Ghahramani, editors, *Advances in Neural Information Processing Systems*, pages 569–576. MIT Press, 2001.

Aharon Ben-Tal, Laurent El Ghaoui, and Arkadi Nemirovski. *Robust Optimization*, volume 28 of *Princeton Series in Applied Mathematics*. Princeton University Press, 2009.

Dimitris Bertsimas and Omid Nohadani. Robust maximum likelihood estimation. *INFORMS Journal on Computing*, 31(3):445–458, 2019.

John M. Danskin. The theory of max-min, with applications. *SIAM Journal on Applied Mathematics*, 14(4):641–664, 1966.

Adnan Darwiche. *Modeling and Reasoning with Bayesian Networks*. Cambridge University Press, 2009.

Gal Elidan and Stephen Gould. Learning bounded treewidth bayesian networks. In Daphne Koller, Dale Schuurmans, Yoshua Bengio, and Léon Bottou, editors, *Advances in Neural Information Processing Systems*, pages 417–424. Curran Associates, Inc., 2008.

Diederik P. Kingma and Max Welling. Auto-encoding variational bayes. In Yoshua Bengio and Yann LeCun, editors, *2nd International Conference on Learning Representations*, 2014.

Doga Kisa, Guy Van den Broeck, Arthur Choi, and Adnan Darwiche. Probabilistic sentential decision diagrams.

In Chitta Baral, Giuseppe De Giacomo, and Thomas Eiter, editors, *Principles of Knowledge Representation and Reasoning: Proceedings of the Fourteenth International Conference, KR 2014, Vienna, Austria, July 20-24, 2014*. AAAI Press, 2014.

Hugo Larochelle and Iain Murray. The neural autoregressive distribution estimator. In Geoffrey J. Gordon, David B. Dunson, and Miroslav Dudík, editors, *AISTATS*, volume 15 of *JMLR Proceedings*, pages 29–37. JMLR.org, 2011.

Yann LeCun and Corinna Cortes. MNIST handwritten digit database. http://yann.lecun.com/exdb/mnist/, 2010.

Daniel Lowd and Jesse Davis. Learning markov network structure with decision trees. In Geoffrey I. Webb, Bing Liu, Chengqi Zhang, Dimitrios Gunopulos, and Xindong Wu, editors, *ICDM 2010, The 10th IEEE International Conference on Data Mining, Sydney, Australia, 14-17 December 2010*, pages 334–343. IEEE Computer Society, 2010.

Denis Deratani Mauá, Diarmaid Conaty, Fabio Gagliardi Cozman, Katja Poppenhaeger, and Cassio Polpo de Campos. Robustifying sum-product networks. *Int. J. Approx. Reason.*, 101:163–180, 2018.

George Papamakarios, Eric Nalisnick, Danilo Jimenez Rezende, Shakir Mohamed, and Balaji Lakshminarayanan. Normalizing flows for probabilistic modeling

and inference. *Journal of Machine Learning Research*, 22(57):1–64, 2021.

Robert Peharz, Robert Gens, Franz Pernkopf, and Pedro M. Domingos. On the latent variable interpretation in sum-product networks. *IEEE Trans. Pattern Anal. Mach. Intell.*, 39(10):2030–2044, 2017.

Robert Peharz, Antonio Vergari, Karl Stelzner, Alejandro Molina, Martin Trapp, Xiaoting Shao, Kristian Kersting, and Zoubin Ghahramani. Random sum-product networks: A simple and effective approach to probabilistic deep learning. In Amir Globerson and Ricardo Silva, editors, *Proceedings of the Thirty-Fifth Conference on Uncertainty in Artificial Intelligence, UAI 2019, Tel Aviv, Israel, July 22-25, 2019*, volume 115 of *Proceedings of Machine Learning Research*, pages 334–344. AUAI Press, 2019.

Robert Peharz, Steven Lang, Antonio Vergari, Karl Stelzner, Alejandro Molina, Martin Trapp, Guy Van den Broeck, Kristian Kersting, and Zoubin Ghahramani. Einsum networks: Fast and scalable learning of tractable probabilistic circuits. In *Proceedings of the 37th International Conference on Machine Learning, ICML 2020, 13-18 July 2020, Virtual Event*, volume 119 of *Proceedings of Machine Learning Research*, pages 7563–7574. PMLR, 2020.

Hoifung Poon and Pedro M. Domingos. Sum-product networks: A new deep architecture. In Fábio Gagliardi Cozman and Avi Pfeffer, editors, *UAI 2011, Proceedings of the Twenty-Seventh Conference on Uncertainty in Artificial Intelligence, Barcelona, Spain, July 14-17, 2011*, pages 337–346. AUAI Press, 2011.

Tahrima Rahman and Vibhav Gogate. Learning ensembles of cutset networks. In Dale Schuurmans and Michael P. Wellman, editors, *Proceedings of the Thirtieth AAAI Conference on Artificial Intelligence, February 12-17, 2016, Phoenix, Arizona, USA*, pages 3301–3307. AAAI Press, 2016.

Tahrima Rahman, Prasanna Kothalkar, and Vibhav Gogate. Cutset networks: A simple, tractable, and scalable approach for improving the accuracy of chow-liu trees. In Toon Calders, Floriana Esposito, Eyke Hüllermeier, and Rosa Meo, editors, *Machine Learning and Knowledge Discovery in Databases - European Conference, ECML PKDD 2014, Nancy, France, September 15-19, 2014. Proceedings, Part II*, volume 8725 of *Lecture Notes in Computer Science*, pages 630–645. Springer, 2014.

Yujia Shen, Arthur Choi, and Adnan Darwiche. Tractable operations for arithmetic circuits of probabilistic models. In Daniel D. Lee, Masashi Sugiyama, Ulrike von Luxburg, Isabelle Guyon, and Roman Garnett, editors, *Advances in Neural Information Processing Systems 29: Annual Conference on Neural Information Processing Systems 2016, December 5-10, 2016, Barcelona, Spain*, pages 3936–3944, 2016.

Jascha Sohl-Dickstein, Eric A. Weiss, Niru Maheswaranathan, and Surya Ganguli. Deep unsupervised learning using nonequilibrium thermodynamics. In Francis R. Bach and David M. Blei, editors, *ICML*, volume 37 of *JMLR Workshop and Conference Proceedings*, pages 2256–2265. JMLR.org, 2015.

Huan Xu, Constantine Caramanis, and Shie Mannor. Robustness and regularization of support vector machines. *Journal of Machine Learning Research*, 10(51):1485–1510, 2009.
