# OpenReview forum: "Robust Learning of Tractable Probabilistic Models"
_auai.org/UAI/2022/Conference — UAI 2022 Poster_

### Official Review · Reviewer_PgHh · 2022-03-21

**Q2(1) Originality/Novelty:** 2
**Q2(2) Significance/Impact:** 2
**Q2(3) Correctness/Technical Quality:** 3
**Q2(6) Clarity Of Writing:** 4
**Q6 Overall Score:** 5
**Q8 Confidence In Your Score:** 3

**Q1 Summary And Contributions:**

This paper proposes two methodologies to train robust tractable probabilistic models. Namely, the work
establishes two ways of training a sum-product network (SPN) and a cutset network (CN).
The robustification is done solving a max-min optimisation problem over the parameters of the networks
(max part) and over a deterministic uncertainty set (min part).

**Q2 Assessment Of The Paper:**

More detailed information regarding each of these aspects is given below:

**Q2(4) Quality Of Experiments (Optional):**

2: Fair: The experimental evaluation is weak: important baselines are missing, or the results do not adequately support the main claims.

**Q2(5) Reproducibility:**

1: Poor: Key details (e.g., proof sketches, experimental setup) are incomplete/unclear, or key resources (e.g., proofs, code, data) are unavailable.

**Q3 Main Strengths:**

The paper is well written and the ideas well explained and easy to follow.
As far as I know, the idea is original.

**Q4 Main Weakness:**

First drawback: Values for h

The uncertainty set contains 0/1 d-dimensional vectors with at most h entries equal to one. This means that at most h bits in each example are flipped. The values for h used in all the experiments are 1, 3 or 5. Now take for example MNIST data set in which each image is represented by a 784-dimensional binary vector, setting h=5 means that less than 1% of the data is being corrupted. With
such low values for h is not possible to conclude that the proposed models are really robust.

Now, a second fundamental drawback is related to the construction of the adversarial and randomly perturbed test
sets. The construction of these sets involves finding examples with the smallest log-likelihood score with respect
to the standard SPN or CN. Is wrong to test the standard models on these test sets, by construction
is obvious they will under perform.
In the tables, we can see that standard models perform better than the robust models when using the original
test data.

**Q5 Detailed Comments To The Authors:**

The algorithm to solve the inner optimisation problem is not discussed. There's only a mention of being
a greedy local search.  **Could you please elaborate more how the inner optimisation is solved?**

Proposition 2 [Danskin, 1966] Involves min of a sum of logarithms, whereas the regularised
maximum likelihood estimators involve the sum of min of logarithms. **Given this subtle change does
[Danskin, 1966] still applies?**

**Q7 Justification For Your Score:**

The experimental setup is weak and there is not real evidence that the methodologies proposed are able to obtain truly robust models.
Authors don't discuss how to exactly solve the inner minimisation problem which makes the reproducibility hard.

My decision is based mainly on the experimental setup (see Q4). There's no enough evidence to conclude that the methodologies proposed are truly obtaining robust models.

**Q9 Complying With Reviewing Instructions:**

1: Yes.

---

### Official Review · Reviewer_1p8t · 2022-03-25

**Q2(1) Originality/Novelty:** 2
**Q2(2) Significance/Impact:** 4
**Q2(3) Correctness/Technical Quality:** 3
**Q2(6) Clarity Of Writing:** 4
**Q6 Overall Score:** 6
**Q8 Confidence In Your Score:** 4

**Q1 Summary And Contributions:**

The paper proposed an algorithm learning robust TPMs where the learned model is expected to be robust on some corrupted examples. Author demonstrates a concise formulation on the robustness and demonstrate the objective that they want to optimize. Finally, author shows an algorithm to learn the robust TPM with the designed objectives. Authors finally empirically demonstrate the effectiveness of the algorithms on both latent and deterministic TPMs.

**Q2 Assessment Of The Paper:**

More detailed information regarding each of these aspects is given below:

**Q2(4) Quality Of Experiments (Optional):**

3: Good: The experimental evaluation is adequate, and the results convincingly support the main claims.

**Q2(5) Reproducibility:**

4: Excellent: Key resources (e.g., proofs, code, data) are available and key details (e.g., proof sketches, experimental setup) are comprehensively described for competent researchers to confidently and easily reproduce the main results.

**Q3 Main Strengths:**

The paper demonstrated a simple and effective formulation to learn a robust TPM. The design on the objective function is very intuitive and clean.

The evaluations on CNs and SPNs is nicely designed. In the experiment, author demonstrated that latent TPMs show different behavior from the non-latent model when training with the robustness objective. It is quite interesting to see that the overall performance penalty for having a robust model is lower in non-latent model (CNs) compared to the latent model (SPN).

**Q4 Main Weakness:**

The work does not exploit much properties of the TPMs. It seems that the Algorithm can be applied on any model whose likelihood computation is tractable. It would be nice to see discussion on various properties of TPMs can help to learn a more robust model. For example CNs has more property than SPN, does this property explain the smaller performance penalty on having a robust model that we see in the evaluation?

As the work focus on robustness of TPM, it would make the work stronger if it can demonstrate the performance of more baseline approaches. For example, how does Algorithm 1 compared to training TPMs with example augmentations, where we add training examples which are perturbed from the original using the uncertainty set?

**Q5 Detailed Comments To The Authors:**

In algorithm 1, author uses a local search to solve the inner optimization problem. As it appears to me that the approximate the inner approximation produce would comprise the quality of the found solution,  it would be nice if author could demonstrate that trade-off between efficiency and quality by using this relaxation.

Has author also considered properties of TPMs to make the inner optimization tractable? For example, if the TPM follows some variable structure, e.g. probabilistic decision graphs (PDGs) or probabilistic sentential decision diagrams (PSDDs), is it possible that we can find the least favorable example tractably? In particular, 1) For example training example x, we can first construct a PDG or PSDD that only assign score 1.0 on examples that are h hamming distance away from x. This circuit should have size O(d * h). Then, we can use the multiply operation on PDGs and PSDDs to obtain a circuit which only assign positive scores on examples that we are interested. At last, the tractable operation that looks for the least probably example on the resulting circuit can identify the least favorable example that are h distance away from x.

**Q7 Justification For Your Score:**

This is the first work that studies the robustness on TPMs, and it shows interesting robustness result on two types of the tractable circuits, CNs and SPNs.

**Q9 Complying With Reviewing Instructions:**

1: Yes.

---

### Official Review · Reviewer_cQGk · 2022-04-03

**Q2(1) Originality/Novelty:** 4
**Q2(2) Significance/Impact:** 3
**Q2(3) Correctness/Technical Quality:** 4
**Q2(6) Clarity Of Writing:** 4
**Q6 Overall Score:** 8
**Q8 Confidence In Your Score:** 4

**Q1 Summary And Contributions:**

The authors propose an approach to learn the parameters of tractable probabilistic models
using robust maximum likelihood estimation. They show that, with their approach, the tested
models perform better under noisy and corrupted data than the same models trained by
simply maximizing the likelihood.

**Q2 Assessment Of The Paper:**

More detailed information regarding each of these aspects is given below:

**Q2(4) Quality Of Experiments (Optional):**

4: Excellent: The experimental evaluation is comprehensive and the results are compelling.

**Q2(5) Reproducibility:**

2: Fair: Key resources (e.g., proofs, code, data) are unavailable but key details (e.g., proof sketches, experimental setup) are sufficiently well-described for an expert to confidently reproduce the main results.

**Q3 Main Strengths:**

The paper is well organized and well written, the proposed approach is interesting and
extensively tested, the results seem convincing.

**Q4 Main Weakness:**

None in particular, the paper is clear and well written.

**Q5 Detailed Comments To The Authors:**

The paper clearly advances the state of the art of tractable probabilistic models by introducing robust parameter learning.
There is not much literature on this topic, so the originality is high.
The impact is good, as the problem of handling noisy or adversarially corrupted data in parameter learning is of high interest in general and in the TPM community in particular.
Technically the paper appears sound.
The experiments are extensive and provide convincing evidence of the thesis of the paper.
Reproducibility is fair since there is no indication that the code will be made public.
The paper is overall clear.

I've some minor suggestions regarding the structure of the paper.

Footnote 1 is important and should be included in the body of the paper and discussed.
In footnote 1, as well as in the references, the names of the authors of the paper
“Robustifying sum-product networks” should be checked: 'Mauá' instead of 'Mau'a' and
'Fabio' instead of 'F'abio'

A full stop is missing at the end of “Distributionally Robust Estimators” point in page 3.

In 3.1: “an uncertainty set defines a boundary or region (of assignments) that are” -> “that is”.

There is an extra space at line 6 of algorithm 1: ‘( see’.

A ‘)’ is missing in the last paragraph before 3.3, after ‘(see Proposition 2’.

At the end of page 5 left column I suggest adding a link to the DeeProb Kit library
source/repository if available.

Page 6 right column: ‘there is several orders’ -> ‘there are several orders’.

"In particular, there
is an order of magnitude difference in the likelihood scores
of SPN−a and SPN−r for h = 5.": this sentence is not clear if you don't say explicitly what is the base
of the logarithm. If it is e, an order of magnitude difference is 2.3, there does not seem to be such a
difference for all datasets. Moreover, it is not clear what is the baseline to which you measure the
difference: is it the case for h=1 or SPN?


**Q7 Justification For Your Score:**

The paper is clear, well written, and presents an interesting approach to handle corrupted
data (a common scenario in real-world situations). The proposed method is extensively
tested on many datasets.

**Q9 Complying With Reviewing Instructions:**

1: Yes.

---

### Official Review · Reviewer_did1 · 2022-04-14

**Q2(1) Originality/Novelty:** 3
**Q2(2) Significance/Impact:** 2
**Q2(3) Correctness/Technical Quality:** 3
**Q2(6) Clarity Of Writing:** 3
**Q6 Overall Score:** 6
**Q8 Confidence In Your Score:** 3

**Q1 Summary And Contributions:**

This paper suggests a new learning method for probabilistic circuits (PCs). This method is based on maximum likelihood estimation (MLE) but it includes a robustness component. MLE expects the dataset to be without corruption, which is non-realistic. Thus, a robust version of MLE is a corruption-free learning method for PCs.

**Q2 Assessment Of The Paper:**

More detailed information regarding each of these aspects is given below:

**Q2(4) Quality Of Experiments (Optional):**

3: Good: The experimental evaluation is adequate, and the results convincingly support the main claims.

**Q2(5) Reproducibility:**

3: Good: Key resources (e.g., proofs, code, data) are available and key details (e.g., proofs, experimental setup) are sufficiently well-described for competent researchers to confidently reproduce the main results.

**Q3 Main Strengths:**

The main strength of this paper is its contribution to a formal definition and treatment of robustness in tractable probabilistic models (TPMs).

Propositions 1 and 2 formally allow for considering data corruption when training cutset networks (CNs), a type of PC. Furthermore, the manuscript emphasizes the use of hyperparameters for controlling this corruption consideration, which is of practical value.

Experimental results are encouraging. In specific Tables 1 and 2 show favourable results in all considered datasets. Moreover, the experiment setup seems extensive, as it involves having an adversarial and randomly perturbed dataset.

**Q4 Main Weakness:**

Few improvements in the theoretical result's presentation can improve the work.

Proposition 1 highlights that the concave problem in Eq (2) is not smooth. A discussion on the practical implications of this consequence would make the paper's contributions stronger. That is because the manuscript claims, in Section 2.1, that Algorithm 1 "can be easily applied to other tractable models". Thus, how could the lack of the smooth property affect inference in other relevant TPMs?

In Section 3.3, the paper says that Eq (3) can be used in Algorithm 1. I suggest adding a comment regarding how this affects the derivations given in "Practical considerations", such as time complexity and local/global optima analysis.

**Q5 Detailed Comments To The Authors:**

Details are given above in Q3 and Q4.

**Q7 Justification For Your Score:**

Although the relevance of the contributions here could be better clarified in the manuscript, the results are clearly exposed and the experiments are encouraging.

**Q9 Complying With Reviewing Instructions:**

1: Yes.

---

### Decision · Program_Chairs · 2022-05-15

**Decision:**

Accept (Poster)

**Comment:**

Meta Review: The quality of this work is good, and all of the reviewers agree that it should be accepted. The paper addresses a significant problem, provides both sound theoretical and compelling empirical results, and is well written.

Pros
* Formal treatment of robustness, a significant problem
* Algorithms and hyperparameters would be meaningful for realistic settings
* Extensive experiments

Cons
* Some assumptions in the model (such as smoothness) may not be warranted in some cases, such as in the presence of latent variables.
* The approach may not fully exploit the characteristics of the TPM model class.
* Some experiments could be improved, such as adding more noise to training data or avoiding adversarial cases for non-robust approaches.